

# Genome-wide identification of *SHMT* family genes in C$_3$, C$_3$-C$_4$, and C$_4$ Salsoleae s.l. species

Peng Peng[1,2,3,*], Qian Qin[4,5,*], Guzailinuer Kuerban[6], Ting Peng[5,6], Mao Wang[1] and Zhibin Wen[3,5,7]

[1] Xinjiang Key Laboratory for Ecological Adaptation and Evolution of Extreme Environment Biology, College of Life Sciences, Xinjiang Agricultural University, Urumqi, China
[2] State Key Laboratory of Ecological Safety and Sustainable Development in Arid Lands, Xinjiang Institute of Ecology and Geography, Chinese Academy of Sciences, Urumqi, China
[3] China-Tajikistan Belt and Road Joint Laboratory on Biodiversity Conservation and Sustainable Use, Xinjiang Institute of Ecology and Geography, Chinese Academy of Sciences, Urumqi, China
[4] College of Life and Geographic Sciences, Kashi University, Kashi, China
[5] Xinjiang Key Lab of Conservation and Utilization of Biological Resources, Urumqi, China
[6] University of Chinese Academy of Sciences, Beijing, China
[7] The Specimen Museum of Xinjiang Institute of Ecology and Geography, Chinese Academy of Sciences, Urumqi, China
* These authors contributed equally to this work.

Corresponding authors
Mao Wang, wangmao88@126.com
Zhibin Wen,
zhibinwen@ms.xjb.ac.cn

## ABSTRACT

C$_4$ photosynthesis is a carbon-concentrating mechanism that evolved to enhance photosynthetic efficiency under conditions favoring photorespiration, such as high temperature, low atmospheric CO$_2$, and aridity. Photorespiration is considered the primary driving force on the evolution of C$_4$ photosynthesis. Serine hydroxymethyltransferase (SHMT) plays a crucial role in one-carbon metabolism and photorespiration. However, there is a lack of comprehensive bioinformatics investigation on the *SHMT* gene family across different photosynthetic types, specifically comparing C$_3$, C$_4$, and C$_3$-C$_4$ intermediate species. In this study, we conducted a systematic analysis of the *SHMT* gene family regarding gene structure, phylogenetic relationships, expression patterns, and cis-acting element in four Salsoleae species, including C$_3$ species *Salsola junatovii*, C$_3$-C$_4$ intermediate species *Oreosalsola laricifolia*, and two C$_4$ species *Xylosalsola arbuscula* and *Soda foliosa*. The results indicated that 4–5 *SHMT* members were identified in these four species. No fragment duplication were identified, which may explain the lower number of *SHMT* members in each Salsoleae species. The range of exon numbers varied from 4 to 15. Phylogenetic analysis showed that the *SHMTs* from Salsoleae species can be classified into four distinct classes, with most members displaying conserved gene structure and motif numbers, except for *OlSHMT3* and *XaSHMT3*, which had divergent gene structures. The *SHMTs* in Salsoleae species did not exhibit organ-specific expression patterns; however, variations in expression were observed among the different members. Analysis of newly sequenced Salsoleae transcriptomes data and published data from five other genera (*Flaveria*, *Heliotropium*, *Mollugo*, *Alternanthera*, and *Neurachne*) revealed that, compared to C$_3$ and C$_3$-C$_4$ intermediate species, only mitochondrial-localized, leaf preferential *SHMT1* showed a low expression among *SHMT* members, probably evolved in C$_4$ photosynthesis evolution. The MYB transcription factors were predicted to be the most significant regulators of *SHMT1*

in three Salsoleae species and the second most significant in *X. arbuscula*. These results may provide valuable information for further analyses, particularly in the evolutionary study of Salsoleae *SHMT1*.

# INTRODUCTION

Photorespiration is a consequence of the dual affinity of the enzyme ribulose −1, 5-bisphosphate carboxylase/oxygenase (Rubisco) for both $CO_2$ and $O_2$. The carboxylation reaction catalyzed by Rubisco produces two molecules of 3-phosphoglycerate (3-PGA), which can be reconverted into ribulose bisphosphate (RuBP) *via* the Calvin cycle. In contrast, the oxygenation reaction yields one molecule of 3-PGA and one molecule of 2-phosphoglycolate (2-PG). Given that 2-PG is toxic to plants, its conversion back to 3-PGA occurs through a reaction sequence known as the photorespiratory carbon cycle (*Bowes, Ogren & Hageman, 1971*; *Voll et al., 2006*; *Bauwe, Hagemann & Fernie, 2010*; *Bräutigam & Gowik, 2016*). This process consumes ATP and NADPH, ultimately resulting in a net release of $CO_2$ from the plant. Under hot and dry conditions, photorespiration can reduce the efficiency of carbon fixation in plants by as much as 30% (*Bauwe, Hagemann & Fernie, 2010*; *Raines, 2011*). Furthermore, aside from its primary role in the photorespiratory carbon cycle, this pathway may also serve secondary functions, such as contributing to the synthesis of glycine and serine (*Wingler et al., 2000*) or playing a role in pathogen defense (*Foyer et al., 2009*).

$C_4$ plants evolved independently approximately 61 times from $C_3$ ancestors, involved the modifications in leaf anatomy, physiology and gene expression (*Sage, 2017*). Compared with $C_3$ plants, the most well-documented $C_4$ plants develop Kranz anatomy, where the outer layer consists of mesophyll cells that fix atmospheric $CO_2$ in the $C_4$ cycle, and the inner layer comprises bundle sheath cells that facilitate the effective donation of $CO_2$ from the decarboxylation of $C_4$ acids to Rubisco, thereby minimizing competition with $O_2$ and reducing photorespiration (*Edwards & Voznesenskaya, 2011*). $C_3$-$C_4$ intermediates ($C_2$ species) are characterized by a photorespiration pump, which restricts the activity of the glycine decarboxylase complex (GDC) to the bundle sheath cells (*Schulze, Westhoff & Gowik, 2016*). These intermediates are considered as evolutionary stepping stones toward $C_4$ photosynthesis based on the current model of $C_4$ evolution (*Sage, Khoshravesh & Sage, 2014*; *Bräutigam & Gowik, 2016*; *Lundgren, 2020*; *Schlüter & Weber, 2020*). Photorespiration is regarded as a major driving force on $C_4$ evolution (*Bräutigam & Gowik, 2016*). In comparison to $C_3$ plants, there is a notable decrease in the expression of photorespiratory genes in $C_4$ plants, particularly the core enzymes of the photorespiration pathway, while the transcript and protein levels in $C_2$ plants remain constant or even higher compared to $C_3$ plants (*Mallmann et al., 2014*; *Lauterbach et al., 2017*; *Siadjeu, Lauterbach & Kadereit, 2021*; *Lauterbach et al., 2024*).

Serine hydroxymethyltransferase (SHMT, EC 2.1.2.1), which depends on pyridoxal 5′-phosphate, is one of the eight core enzymes in the canonical photorespiratory pathway (*Hagemann et al., 2016*). SHMT plays a catalytic role in the transformation between serine and glycine with GDC (*Hanson, Gage & Shachar-Hill, 2000*), as well as in the synthesis of tetrahydrofolate (H4PteGlun, THF)/5,10-methylenetetrahydrofolate (5,10-CH2-H4PteGlun), the synthesis of methionine, and the maintenance of redox balance during photorespiration (*Schirch, 1982*; *Appaji Rao et al., 2003*; *Zhang et al., 2010*). It is widely distributed across plants, animals, and microorganisms (*Prabhu et al., 1996*; *Hanson, Gage & Shachar-Hill, 2000*). Members of the *SHMT* gene family have been reported in many species, including *Arabidopsis*, soybean (*Lakhssassi et al., 2019*), rice (*Pan et al., 2024*), cucumber (*Gao et al., 2022*), tomato (*Liu et al., 2022*), and alfalfa (*Gao et al., 2024*). The *SHMT* gene family members range from five (rice, *Pan et al., 2024*) to eighteen (soybean, *Lakhssassi et al., 2019*). Depending on their subcellular localization, there are four kinds of *SHMTs*, distributed in mitochondria, chloroplast, cytoplasm, and the nucleus, respectively (*Zhang et al., 2010*; *Nogués et al., 2022*), indicating their diverse roles in metabolic pathways (*Voll et al., 2006*; *Hagemann et al., 2016*; *Lakhssassi et al., 2019*; *Liu et al., 2022*; *Gao et al., 2024*). Currently, mitochondrial-localized *SHMTs* has been extensively studied and are known to participate in the process of photorespiration, one carbon metabolism, plant growth, and stress response (*Voll et al., 2006*; *Liu et al., 2022*; *Yuan et al., 2022*). In *Arabidopsis*, there are two mitochondrial *SHMTs*, namely *AtSHM1* and *AtSHM2*. *AtSHM1* is predominantly expressed in leaves, whereas *AtSHM2* is mainly expressed in shoot and roots (*Voll et al., 2006*). Only *AtSHM1* is involved in the photorespiratory carbon cycle, and the mutation of this gene causes a photorespiratory phenotype in *Arabidopsis thaliana* (*Voll et al., 2006*). Additionally, *AtSHM1* plays a regulatory role in sucrose accumulation and the homeostasis of reactive oxygen species (ROS), both of which are crucial for primary root growth (*Yuan et al., 2022*). Mitochondrial *OsSHMT1* from rice and *GmSHMT08* from soybean are involved in defense mechanisms against abiotic and biotic stress (*Wang et al., 2015*; *Lakhssassi et al., 2020*). Furthermore, the mitochondrial *SlSHMT* from tomato interacts with chaperonin 60α1 (SlCPN60α1) to regulate photosynthesis and photorespiration processes (*Ye et al., 2020*).

The family Chenopodiaceae s.s. (Amaranthaceae s.l.), as classified by APG IV (2016) comprises approximately 558 species, making it the third largest group of $C_4$ species. The tribe Salsoleae encompasses over half of the known $C_4$ species (310) within Chenopodiaceae s.s. (*Sage, 2017*), and it also includes $C_3$ and $C_3$-$C_4$ species (*Voznesenskaya, 2001*; *Wen & Zhang, 2015*). The diversity in habitats, life forms and photosynthetic characteristics in the assimilation organs of Salsoleae is particularly complex (*Edwards & Voznesenskaya, 2011*). Unlike most $C_4$ lineages dominated by herbaceous species, these $C_4$ Salsoleae species also include subshrubs and shrubs, and even rarely small trees (*Zhu, Mosyankin & Clemants, 2003*). Furthermore, many $C_4$ Chenopodiaceae s.s. plants frequently dominate warm temperate and tropical grasslands and savannas, particularly in environments such as sand dunes, salt marshes, semideserts, and deserts (*Kadereit et al., 2003*).

Due to the limited number of whole genomes sequenced to date, no comparisons of *SHMTs* across different photosynthetic species have been made to elucidate the genetic evolution and function of *SHMTs*. This study investigates the genome-wide identification and characterization of *SHMT* genes in four species from Salsoleae. *Salsola junatovii* ($C_3$ species), *Oreosalsola laricifolia* ($C_2$ species), *Xylosalsola arbuscula* ($C_4$ species), and *Soda foliosa* ($C_4$ species). A systematic analysis was conducted on gene family number, gene structures, conserved motifs, evolutionary relationships, collinear relationships, cis-acting element distributions, and tissue patterns. Additionally, the published leaf transcriptome data from five genera, including the dicots *Flaveria*, *Heliotropium*, *Mollugo*, *Alternanthera*, as well as the monocot *Neurachne*, were utilized. These genera encompass various photosynthetic species to assess *SHMTs* gene expression along the emergence of $C_4$ species.

## MATERIALS AND METHODS

### Genome-wide identification of *SHMT* gene family members

In this study, we utilized genome sequencing data for *Salsola junatovii*, *Oreosalsola laricifolia*, *Soda foliosa*, and *Xylosalsola arbuscula*, provided as unpublished genome data by Institute of Genetics and Developmental Biology, Chinese Academy of Sciences. We extracted protein and coding sequences (CDS) for these four Salsoleae species using the 'GXF Sequence Extract' and 'Batch Translate CDS' modules in TBtools (v2.154) (*Chen et al., 2023*). Reference protein sequences of *Arabidopsis thaliana SHMTs* were obtained from the TAIR database (https://www.arabidopsis.org/), and the Hidden Markov Model (HMM) of the SHMT conserved domain (PF00464) was downloaded from the Pfam database (https://www.ebi.ac.uk/interpro/entry/pfam/). Using the SHMT HMM model as a template, we employed HMMER (v3.0) to perform a whole-genome scan of these Salsoleae species to identify potential *SHMTs*. To validate these candidates, we constructed a local protein database for each Salsoleae species using BLAST (v2.14), with *A. thaliana* SHMT protein sequences as queries (E-value threshold set at 1e−5) (*Ahmad et al., 2024*). The candidate genes were further refined by integrating results from HMMER and BLAST, and protein sequences were extracted using the 'Fasta Extract' module in TBtools. All candidate genes were confirmed for structural integrity using the NCBI Conserved Domains Database (https://www.ncbi.nlm.nih.gov/Structure/cdd/wrpsb.cgi). The *SHMT* numbers of these Salsoleae species were designated according to homologous genes in *A. thaliana*. Finally, molecular weight (Mw) and isoelectric point (pI) predictions for all SHMT protein-coding genes were conducted using the ExPASy online tool (https://web.expasy.org/compute_pi/) (*Wilkins et al., 1999*).

### SHMT protein and gene structure analysis

Secondary structure predictions for SHMT proteins were performed using the SOPMA online tool (https://npsa.lyon.inserm.fr/cgi-bin/npsa_automat.pl?page=/NPSA/npsa_sopma.html). The three-dimensional structure of SHMT was predicted utilizing AlphaFold3 (https://alphafoldserver.com/), and the resulting models were simulated and visualized with PyMOL (v3.1.0) (*Jumper et al., 2021*; *Mooers, 2020*). The subcellular

localization of SHMT proteins was inferred through the CELLO online tool (http://cello.life.nctu.edu.tw/) (*Yu et al., 2006*), while signal peptide predictions were performed using SignalP (v5.0) (https://services.healthtech.dtu.dk/services/SignalP-5.0/) (*Nielsen et al., 2019*).

For gene structure and conserved motif analysis, the locations of *SHMT* genes and their exon-intron structures in these species were extracted from the General Feature Format (GFF) annotation files and visualized using the 'Gene Location Visualize from GTF/GFF' function in TBtools. Conserved motifs within SHMT protein sequences were identified using MEME (https://meme-suite.org/meme/tools/meme), with the motif number maintained at 20 and other parameters set to default (*Bailey et al., 2015*). These results were visualized using the 'Gene Structure View' function in TBtools.

## Phylogenetic analysis

SHMT protein sequences from *A. thaliana*, *Glycine max*, *Solanum lycopersicum*, *Populus trichocarpa*, *Cucumis sativus*, *Beta vulgaris* and *Oryza sativa* were retrieved from Phytozome (v13) and CuGenDB (http://cucurbitgenomics.org/). These sequences were aligned using ClustalW, with the Delay Divergent Cutoff (%) set to 30 (*Liu et al., 2022*), while all other options remained at their default settings. A phylogenetic tree comprising 71 SHMT protein sequences was constructed using the maximum likelihood method in MEGA-X (v10.1.8), applying the Jones-Taylor-Thornton (JTT) amino acid substitution model with uniform rates among sites (no discrete gamma categories or invariant sites), the Nearest-Neighbor-Interchange (NNI) heuristic search (initial tree generated automatically by NJ/BioNJ), and 1,000 bootstrap replicates; all other parameters were left at their defaults (*Gao et al., 2024*). The resulting phylogenetic tree was subsequently visualized using iTOL (v7.0) (https://itol.embl.de/).

## *SHMT* gene family syntenetic analysis

The gene location information for *SHMT* family members in four Salsoleae species was analyzed using the 'Gene Location Visualize from GTF/GFF' function in TBtools (*Chen et al., 2023*). Gene density was calculated utilizing the 'Gene Density Profile' function. Synteny within each species was assessed using the 'One Step MCScanX' module in TBtools, with results visualized through the 'Advanced Circos' function. Furthermore, synteny relationships among these species, *A. thaliana*, and *B. vulgaris* were examined by downloading the genomic data of *Arabidopsis* and *B. vulgaris* from Phytozome (v13) and employing the 'One Step MCScanX' Synteny analysis plots were generated to illustrate the syntenic relationships of homologous *SHMT* genes across these species.

## RNA extraction and reverse transcription-qPCR

Seeds from *S. junatovii*, *O. laricifolia*, *X. arbuscula*, and *S. foliosa* were collected in October 2023 in Xinjiang. The seeds were air-dried at room temperature for 2 weeks and subsequently stored at 4 °C in a refrigerator. Following sterilization (*Guo et al., 2024*), the seeds were sown on 1/2 MS solid medium (1/2 MS + 15 g L$^{-1}$ sucrose + 8 g L$^{-1}$ agar) and incubated in a growth chamber for 3 days. The growth chamber was maintained under a

14 h light/10 h dark cycle at 25 °C during the light period and 15 °C during the dark period. The light intensity was maintained at approximately 300 µmol·m$^{-2}$·s$^{-1}$. Subsequently, healthy seedlings were selected and transferred to a Hoagland nutrient solution for hydroponic culture, with the solution being changed every 3 days. After 6 to 8 weeks of growth, leaf tissue samples were collected at 11:00 AM, immediately frozen in liquid nitrogen, and stored at −80 °C for subsequent total RNA extraction. Root, stem, and leaf tissue samples were collected from these Salsoleae species, with each tissue sample weighing approximately 100 mg. The samples were immediately frozen in liquid nitrogen and ground into a fine powder.

Total RNA was extracted using the TaKaRa MiniBEST Plant RNA Extraction Kit. The quality and concentration of the RNA were assessed using a NanoDrop spectrophotometer, and only RNA samples with a 260/280 ratio between 1.8 and 2.1 were selected for further analysis. To eliminate DNA contamination from the samples and synthesize complementary DNA (cDNA), we employed the PrimeScript™ RT Reagent Kit with gDNA Eraser. Reverse transcription- qPCR (RT-qPCR) primers specific to the *SHMTs* in these species were designed using the NCBI Primer Design Tool (https://www.ncbi.nlm.nih.gov/tools/primer-blast/index.cgi). The designed primers were submitted to Sangon Biotech for synthesis *via* the PAGE purification method. Quantitative RT-PCR was performed utilizing species-specific *β-actin* genes as internal reference controls for normalization (*Zhang et al., 2019b*), with the primer sequences for both *SHMT* and *β-actin* genes in each species provided (Table S1). RT-qPCR experiments were conducted with the TB Green *Premix Ex Taq*™ II kit. All experiments were carried out in the molecular laboratory of the research group at the Xinjiang Institute of Ecology and Geography Chinese Academy of Sciences (*Zhang et al., 2019a*). For each species, the average expression level of all gene members in the root tissue was used as the control, and the relative expression levels of *SHMTs* in the root, stem, and leaf tissues were calculated using the $2^{-\Delta\Delta CT}$ method (*Livak & Schmittgen, 2001*). The expression data were log-transformed (log10), and the results were visualized using the ggplot2 package (v3.5.1).

## Gene expression level analysis based on RNA-seq data

To further investigate the expression patterns of *SHMT* family members associated with different photosynthetic types (C$_2$, C$_3$, and C$_4$) across various plant groups, we selected five plant groups that include species exhibiting C$_2$, C$_3$, and C$_4$ photosynthesis based on prior studies. Subsequently, we downloaded 27 RNA-seq datasets corresponding to each of the 27 species for further analysis. These datasets comprise dicotyledonous plants from *Alternanthera* (Amaranthaceae), *Flaveria* (Asteraceae), *Heliotropium* (Boraginaceae), and *Mollugo* (Molluginaceae), as well as monocotyledonous plants from *Neurachne* (Poaceae) (*Chinthapalli et al., 2000*; *Voznesenskaya et al., 2013*; *Stata et al., 2014*; *Wen & Zhang, 2015*; *Tao, Lyu & Zhu, 2016*; *Thulin et al., 2016*; *Lundgren, 2020*; *Lyu et al., 2020*; *Bernardo et al., 2023*; *Lauterbach et al., 2024*). Additionally, newly sequenced RNA-seq data from four species of Salsoleae (Amaranthaceae) were included (Table S2). All RNA-seq data were generated using a paired-end sequencing strategy and underwent quality control using

FastQC. The transcriptomes were assembled *de novo* using Trinity (v2.11.0) with default parameters (*Haas et al., 2013*). Subsequently, Cluster Database at High Identity with Tolerance (CD-HIT) (v4.8.1) was employed to cluster the transcript sequences at a 0.95 similarity threshold, effectively removing redundant transcripts (*Fu et al., 2012*). To assess the quality of the transcript assembly, Bowtie2 (v2.4.4) was utilized to map the reads back to their respective transcriptomes (*Langmead & Salzberg, 2012*), while Salmon (v1.10.3) was used for the quantitative analysis of the assembled transcripts (*Patro et al., 2017*). The assembled 31 transcripts were annotated using the *Arabidopsis* database (https://www.arabidopsis.org/), and the transcips per million (TPM) values of the *SHMT* gene family members for each species were extracted to measure transcript abundance. Finally, stacked bar plots were generated using the ggplot2 package (v3.5.1) to display the expression levels of *SHMT* in Salsoleae plants and the five aforementioned genera. Individual bar plots were created for each *SHMT* member to illustrate the expression of *SHMTs* in plants with different photosynthetic types.

## Prediction of cis-acting elements and transcription factor binding sites in promoter sequences

Prediction of cis-acting elements and transcription factor binding sites in promoter sequences located 2,000 bp upstream of the *SHMT* coding sequence (CDS) were extracted using the GTF/GFF3 Sequences Extract function in TBtools. These sequences were then submitted to the PlantCARE website (https://bioinformatics.psb.ugent.be/webtools/plantcare/html/) for predicting cis-acting elements (*Lescot, 2002*), while transcription factor binding sites were predicted utilizing the PlantTFDB database (https://planttfdb.gao-lab.org/) (*Tian et al., 2020*). The results were systematically organized and visualized through the 'Gene Structure View (Advanced)' and 'Heatmap' functions available in TBtools.

## RESULTS

### Genome-wide identification of *SHMT* gene family member

Based on the whole genome data of four Salsoleae species, we identified four to five members of each *SHMT* gene family. *O. laricifolia* and *X. arbuscula* each contained five *SHMTs*, including *SHMT1*, *SHMT2*, *SHMT3*, *SHMT4*, and *SHMT7*. In contrast, *S. junatovii* and *S. foliosa* each possessed four members, namely *SHMT1*, *SHMT2*, *SHMT4*, and *SHMT7* (Table 1). All identified *SHMTs* contained the characteristic SHMT domain (Pfam: PF00464) (Table S3). Further analysis of the physicochemical properties of these *SHMT* members (Table 1) revealed that the amino acid lengths ranged from 458 (*OlSHMT3*) to 615 (*SjSHMT7*, *OlSHMT7*, *SfSHMT7*, and *XaSHMT7*). Notably, *SHMT7* across these species exhibited the same amino acid length. Among these members, *XaSHMT7* had the largest molecular weight at 68,405.64 Da, while *OlSHMT3* had the smallest molecular weight at 50,244.21 Da. Additionally, the isoelectric points (pI) of these *SHMTs* ranged from 5.98 (*OlSHMT7*) to 8.86 (*XaSHMT4*). It was noteworthy that *SHMT3* and *SHMT7* were classified as acidic proteins (pI < 7).

**Table 1 Characteristics of *SHMT* gene family members based on four Salsoleae species genomes.**

| Gene name | Gene ID | Gene locus | Amino acids(aa) | Molecular weight (kDa) | pI | SHMT domain location |
|---|---|---|---|---|---|---|
| *SjSHMT1* | *ts3g08640_T02* | Chr3 | 539 | 59,960.89 | 8.07 | 54–475 |
| *SjSHMT2* | *ts2g04227_T01* | Chr2 | 514 | 57,033.97 | 7.18 | 52–449 |
| *SjSHMT4* | *ts2g06369_T02* | Chr2 | 540 | 59,438.62 | 8.59 | 81–481 |
| *SjSHMT7* | *ts3g08322_T02* | Chr3 | 615 | 68,383.58 | 6.08 | 157–562 |
| *OlSHMT1* | *sy2g05957_T04* | Chr2 | 515 | 56,965.33 | 8.38 | 51–451 |
| *OlSHMT2* | *sy1g02520_T01* | Chr1 | 515 | 57,113.13 | 8.59 | 53–450 |
| *OlSHMT3* | *sy8g29094_T01* | Chr8 | 458 | 50,244.21 | 6.17 | 105–402 |
| *OlSHMT4* | *sy1g00078_T01* | Chr1 | 540 | 59,350.55 | 8.47 | 81–481 |
| *OlSHMT7* | *sy2g06541_T03* | Chr2 | 615 | 68,303.52 | 5.98 | 43–515 |
| *SfSHMT1* | *jg1g00713_T01* | Chr1 | 515 | 57,035.35 | 7.70 | 54–451 |
| *SfSHMT2* | *jg2g04576_T01* | Chr2 | 516 | 57,141.09 | 7.18 | 54–451 |
| *SfSHMT4* | *jg2g02805_T02* | Chr2 | 540 | 59,431.62 | 8.47 | 81–481 |
| *SfSHMT7* | *jg1g01220_T02* | Chr1 | 615 | 68,302.56 | 6.03 | 157–562 |
| *XaSHMT1* | *mbsg23959_T02* | Scf7 | 512 | 56,804.11 | 7.67 | 51–448 |
| *XaSHMT2* | *mb1g00716_T01* | Chr2 | 517 | 57,284.28 | 7.67 | 55–452 |
| *XaSHMT3* | *mb5g11984_T01* | Chr**5** | 532 | 58,316.41 | 6.21 | 86–476 |
| *XaSHMT4* | *mb4g10902_T02* | Chr4 | 541 | 59,569.08 | 8.86 | 82–482 |
| *XaSHMT7* | *mb5g13083_T01* | Chr7 | 615 | 68,405.64 | 6.31 | 160–562 |

**Note:**

*Sj, Salsola junatovii; Ol, Oreosalsola laricifolia; Sf, Soda foliosa; Xa, Xylosalsola arbuscula; SHMT, Serine Hydroxymethyltransferase; pI, Isoelectric point.*

## Prediction of SHMT protein secondary structure and subcellular localization

The secondary structures of the SHMT proteins were analyzed (Table 2). The results indicated that the *SHMT* family across all four Salsoleae species was predominantly composed of α-helix, extended strands, β-turns, and random coils, with proportions ranging from 37.4% to 48.05%, 9.27% to 11.87%, 2.15% to 3.93%, and 38.13% to 49.27%, respectively. Subcellular localization analysis revealed that SHMT proteins were primarily distributed in the mitochondria, chloroplasts, and nuclei: *SHMT1* and *SHMT2* in each species were predicted to localize mainly to mitochondria, while *SHMT3* and *SHMT4* were predicted to localize to chloroplasts, and *SHMT7* to nuclei. Signal peptide prediction indicated that none of the *SHMT* members possessed a typical signal peptide region. Additionally, to further analyze their structural characteristics, we predicted and visualized the three-dimensional structures of *SHMT1*, *SHMT2*, *SHMT4*, *SHMT7*, and *SHMT3* (Fig. S1), with confidence scores are provided (Table S4). The results demonstrated that the overall fold and the distribution of the SHMT domain (pfam00464) were highly conserved among all four species, and each SHMT isoform assembled into a homotetrameric structure. However, distinct local structural variations were evident among different

**Table 2 The secondary structure and subcellular localization in *SHMT* members in four Salsoleae species.**

| Gene name | Alpha helix | Extended strand | Beta turn | Random coil | CELLO |
|---|---|---|---|---|---|
| *SjSHMT1* | 48.05% | 10.58% | 2.60% | 38.78% | Mitochondrial |
| *SjSHMT2* | 47.08% | 11.09% | 3.70% | 38.13% | Mitochondrial |
| *SjSHMT4* | 43.52% | 10.19% | 2.78% | 43.52% | Chloroplast |
| *SjSHMT7* | 38.70% | 9.76% | 2.44% | 49.11% | Nuclear |
| *OlSHMT1* | 46.80% | 11.07% | 3.11% | 39.03% | Mitochondrial |
| *OlSHMT2* | 47.96% | 11.07% | 2.52% | 38.45% | Mitochondrial |
| *OlSHMT3* | 46.51% | 10.70% | 3.93% | 38.86% | Chloroplast |
| *OlSHMT4* | 44.07% | 11.30% | 2.41% | 42.22% | Chloroplast |
| *OlSHMT7* | 38.05% | 10.24% | 2.44% | 49.27% | Nuclear |
| *SfSHMT1* | 46.80% | 10.87% | 2.72% | 39.61% | Mitochondrial |
| *SfSHMT2* | 44.77% | 10.27% | 3.49% | 41.47% | Mitochondrial |
| *SfSHMT4* | 44.63% | 11.67% | 2.59% | 41.11% | Chloroplast |
| *SfSHMT7* | 44.07% | 9.27% | 2.28% | 44.39% | Nuclear |
| *XaSHMT1* | 44.73% | 10.55% | 2.15% | 42.58% | Mitochondrial |
| *XaSHMT2* | 46.81% | 10.83% | 2.71% | 39.65% | Mitochondrial |
| *XaSHMT3* | 43.61% | 10.71% | 3.20% | 42.48% | Chloroplast |
| *XaSHMT4* | 46.40% | 10.35% | 2.22% | 41.04% | Chloroplast |
| *XaSHMT7* | 37.40% | 11.87% | 2.60% | 48.13% | Nuclear |

**Note:**
*Sj, Salsola junatovii; Ol, Oreosalsola laricifolia; Sf, Soda foliosa; Xa, Xylosalsola arbuscula; SHMT, Serine Hydroxymethyltransferase.*

*SHMT* members; although the global fold was maintained, these local conformational differences may underlie the substrate-binding specificity or regulatory functions of each isoform.

### Gene structure and conserved motif analysis of *SHMT* members

To further elucidate the functional differences among *SHMTs*, we analyzed their gene structures and conserved motifs. The conserved motif analysis revealed the presence of 20 conserved motifs, with lengths ranging from 8 to 50 amino acids (Table S5). Utilizing TBtools software, we categorized the *SHMTs* into four distinct classes based on phylogenetic analysis and conserved motifs (Fig. 1A). Specifically, *SHMT1* and *SHMT2* from each species were classified into Class IV, while *SHMT3* from *O. laricifolia* and *X. arbuscula* was assigned to Class III. All *SHMT4s* were placed in Class II, and *SHMT7s* were categorized into Class I. Notably, SHMT proteins within the same group exhibited similar motif compositions (Fig. 1A), with the exception of *SHMT3*. Motifs 1, 3, 4, 5, and 8 were present in all *SHMT* members. Furthermore, the number of exons was consistent within each class (Fig. 1B). *SHMTs* in Class I, Class II, Class III, and Class IV contained 4, 4, 10, and 15 exons, respectively. These analyses suggested structural differences among *SHMT* members across different classes, implying potential functional differences.

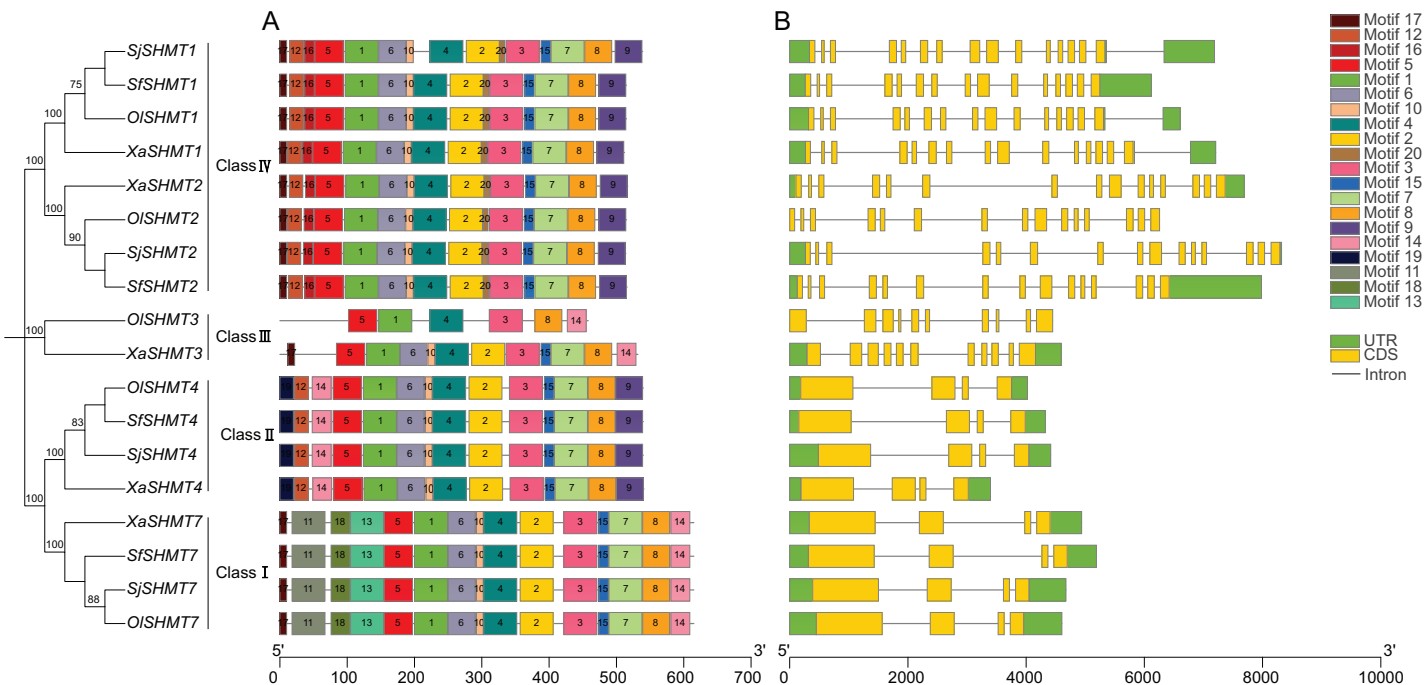

**Figure 1 Conserved motifs (A) and exon-intron structure (B) of the *SHMT* members in four Salsoleae species.** Different colors represent different motifs. The phylogeny tree was constructed based on the full length of SHMT protein sequences using MEGA 7.0. *Sj, Salsola junatovii*; *Ol, Oreosalsola laricifolia, Sf, Soda foliosa*; *Xa, Xylosalsola arbuscula*.

## Phylogenetic analysis of the *SHMT* gene family

To investigate the phylogenetic relationships among members of the *SHMT* gene family, this study selected seven representative species, including six dicot species (*A. thaliana*, *G. max*, *S. lycopersicum*, *B. vulgaris*, *C. sativus*, and *P. trichocarpa*) and one monocot species (*O. sativa*) (Table S6). Based on the topology of the phylogenetic tree constructed using the maximum likelihood method, the 71 *SHMTs* were classified into four classes (Fig. 2). Members of Class I, Class III, and Class IV were localized in the nucleus, chloroplast, and mitochondrion, respectively. Group II exhibited two predicted subcellular localizations: *SHMTs* from *B. vulgaris* and the Salsolee species were localized to the chloroplast, while other *SHMTs* were localized in the cytosol.

## Collinearity analysis among Salsoleae species, *A. thaliana* and *B. vulgaris*

We investigated the gene synteny relationships among the four Salsoleae species, as well as their syntenic relationships with *A. thaliana* and *B. vulgaris*. No segmental or tandem duplication events were observed among the Salsoleae species. Chromosomal localization analysis (Table 1, Fig. S2) revealed that the *SHMTs* of *S. junatovii* and *S. foliosa* were distributed across two chromosomes, while the *SHMTs* of *O. laricifolia* and *X. arbuscula* were distributed on three chromosomes. Furthermore, in *S. junatovii*, *O. laricifolia*, and *S. foliosa*, the *SHMT1* and *SHMT7*, as well as *SHMT2* and *SHMT4*, were each located on

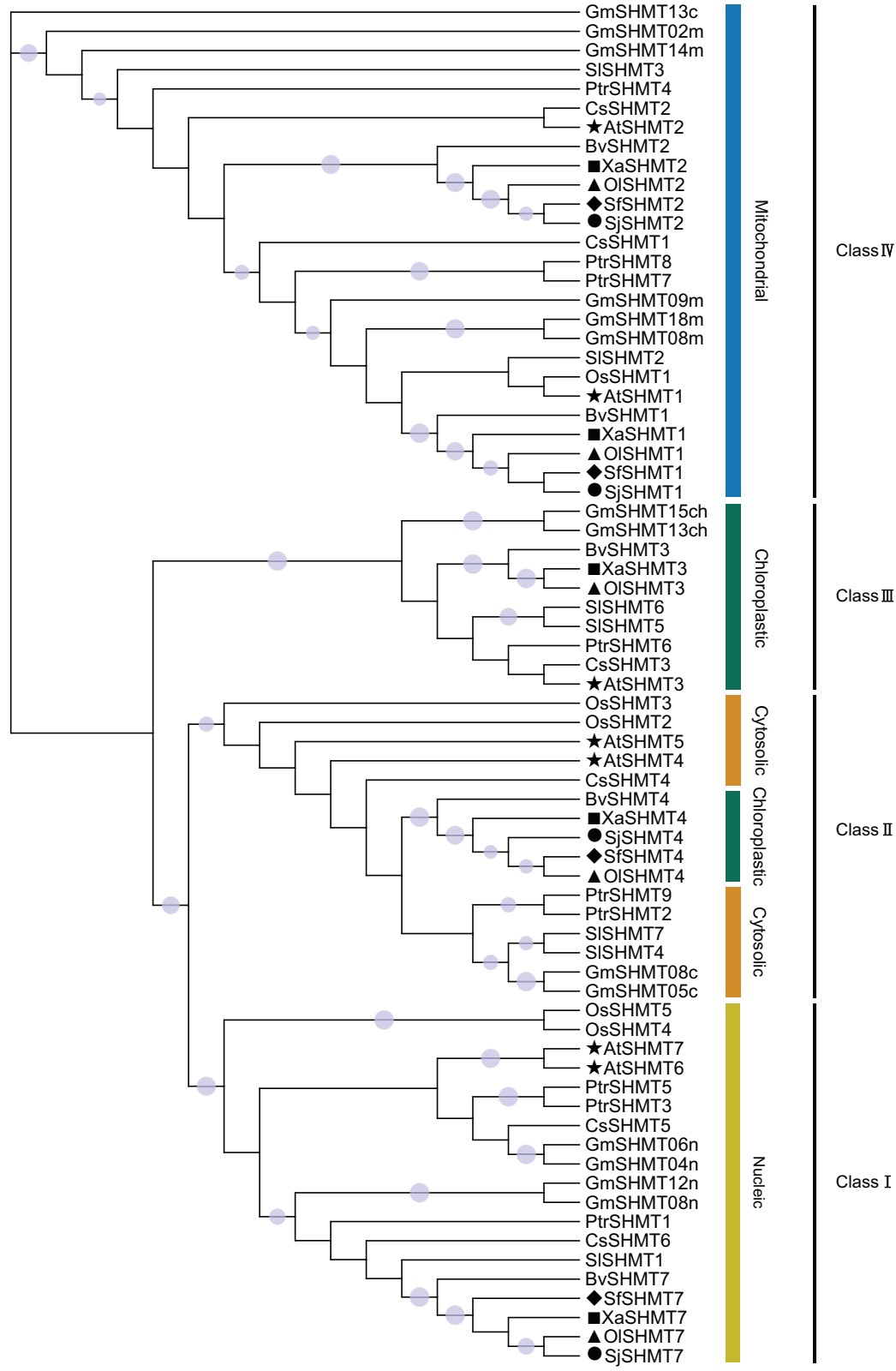

**Figure 2 The unrooted phylogenetic tree of SHMT members based on the maximum likelihood method.** *SHMT* translated protein sequences come from *Arabidopsis thaliana* (7 protein sequences), *Glycine max* (14 protein sequences), *Solanum lycopersicum* (7 protein sequences), *Populus trichocarpa* (9

**Figure 2** (continued)
protein sequences), *Cucumis sativus* (7 protein sequences), *Beta vulgaris* (5 protein sequences), *Oryza sativa* (5 protein sequences), *Salsola junatovii* (4 protein sequences), *Oreosalsola laricifolia* (5 protein sequences), *Soda foliosa* (4 protein sequences), and *Xylosalsola arbuscula* (5 protein sequences). The proteins in *Oreosalsola laricifolia* (*Ol*), *Salsola junatovii* (*Sj*), *Soda foliosa* (*Sf*), *Xylosalsola arbuscula* (*Xa*), and *Arabidopsis thaliana* are marked with triangles, circles, rhombuses, squares, and pentagrams, respectively. The four classes are colored differently. Only bootstrap values above 0.7 are shown in the phylogenetic tree. The purple circles on the branches represent support values, with larger circles indicating higher support.                                   

the same chromosome, respectively. However, in *O. laricifolia*, the *OlSHMT3* gene was located on a separate chromosome. *X. arbuscula* possessed five *SHMTs*, and exhibited a distinct distribution pattern. Specifically, *XaSHMT3* and *XaSHMT7* were located on the same chromosome, while the other genes were distributed across different chromosomes, with one gene (*XaSHMT1*) found in a scaffold region. This suggests that *X. arbuscula* may have undergone unique genomic rearrangements during its evolutionary history.

Through the analysis of the genetic relationships of *SHMT* genes among Salsoleae species, *Arabidopsis*, and *B. vulgaris*, we found that *SHMT1*, *SHMT2*, and *SHMT4* exhibited strong collinearity across the six species. Notably, the collinearity network of *SHMT1* encompassed all analyzed species (Fig. 3 and Table S7), suggesting that *SHMT1* may exhibit a significant degree of functional conservation across different species. Conversely, *SHMT3* and *SHMT7* displayed significant species-specific evolutionary traits. Specifically, *OlSHMT3* from *O. laricifolia* showed collinearity with *SHMT3* from *B. vulgaris*, whereas *XaSHMT3* from *X. arbuscula* completely lacked this conserved collinearity. The *SHMT7* collinearity network was more limited, with a clear homologous relationship observed solely between *S. junatovii* and *B. vulgaris*. Furthermore, *SHMT4* gene pairs were identified across all species, suggesting that this gene might have existed prior to the divergence of the ancestral species. Notably, Salsoleae species exhibited 1–2 pairs of homologous genes with *Arabidopsis*, while there were more homologous genes (2–4 pairs) with *B. vulgaris*. This may be attributed to the close phylogenetic relationship between Salsoleae and *B. vulgaris*, as both belong to the Amaranthaceae family, leading to greater homology in their *SHMTs*.

## Quantitative real-time PCR analysis in different tissue

Quantitative real-time PCR (RT-qPCR) was employed to analyze the expression levels of *SHMTs* in the roots, stems, and leaves of various Salsoleae species (Fig. 4). The results demonstrated that all *SHMTs* were expressed in all three tissues, indicating a lack of strict tissue specificity. Among these *SHMT* members, *SHMT1* exhibited the highest expression levels, with its transcripts predominantly accumulating in the leaves. Additionally, *SHMT4* and *SHMT7* demonstrated significantly higher expression in leaves compared to roots and stems. The expression pattern of *SHMT2* varied across species, in *O. laricifolia* and *S. foliosa*, *SHMT2* expression was higher in roots and stems than in leaves, whereas in *S. junatovii* and *X. arbuscula*, expression levels were elevated in stems and leaves compared to roots. Notably, *SHMT3* was detected only in *O. laricifolia* and *X. arbuscula*, with

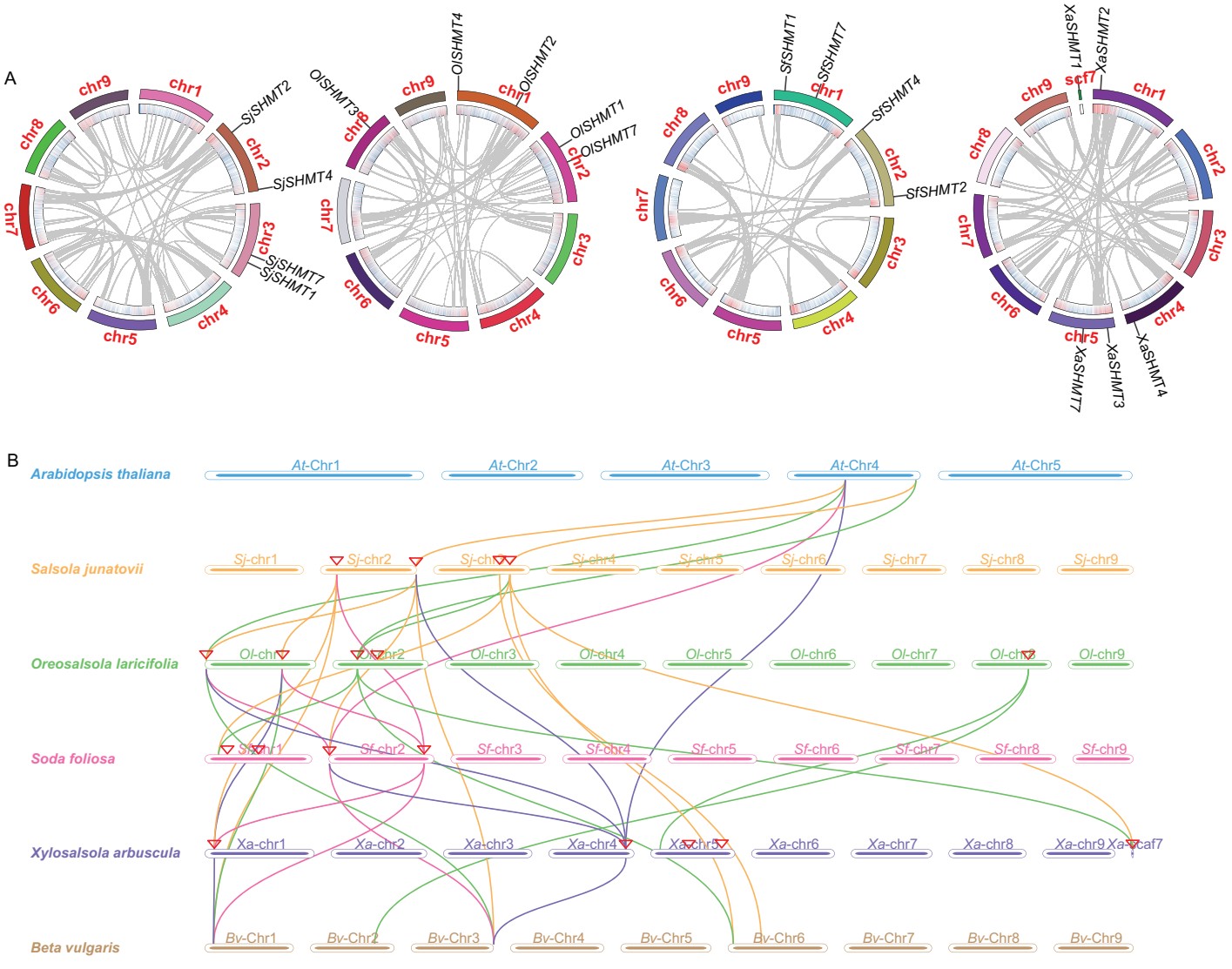

**Figure 3 Gene duplication and collinearity analysis of *SHMTs* in four Salsoleae species.** (A) Schematic representation of the chromosomal distribution and interchromosomal relationships of *SHMTs* in a Circos plot. In the Circos plot, gray lines represent the syntenic relationships within each species genome, the innermost ring indicates gene density, and the outermost ring represents the chromosome numbers of each species. (B) Collinearity analysis of the *SHMT* gene family in *Oreosalsola laricifolia*, *Salsola junatovii*, *Soda foliosa*, *Xylosalsola arbuscula*, *Arabidopsis thaliana*, and *Beta vulgaris*. Different colored lines delineate the syntenic *SHMTs* pairs: orange, green, pink, blue represents syntenic relationships in *Salsola junatovii*, *Oreosalsola laricifolia*, *Soda foliosa*, and *Xylosalsola arbuscula*, respectively.

markedly lower expression compared to other members. These findings suggest that distinct *SHMT* homologs may fulfill specialized functional roles in different organs across Salsoleae species.

## Transcriptional expression analysis of *SHMTs* in different groups

To further investigate the expression of *SHMT* in Salsoleae and other plant groups, we analyzed *SHMTs* across four Salsoleae species and five additional genera: *Alternanthera* (Amaranthaceae), *Flaveria* (Asteraceae), *Heliotropium* (Boraginaceae), *Mollugo*

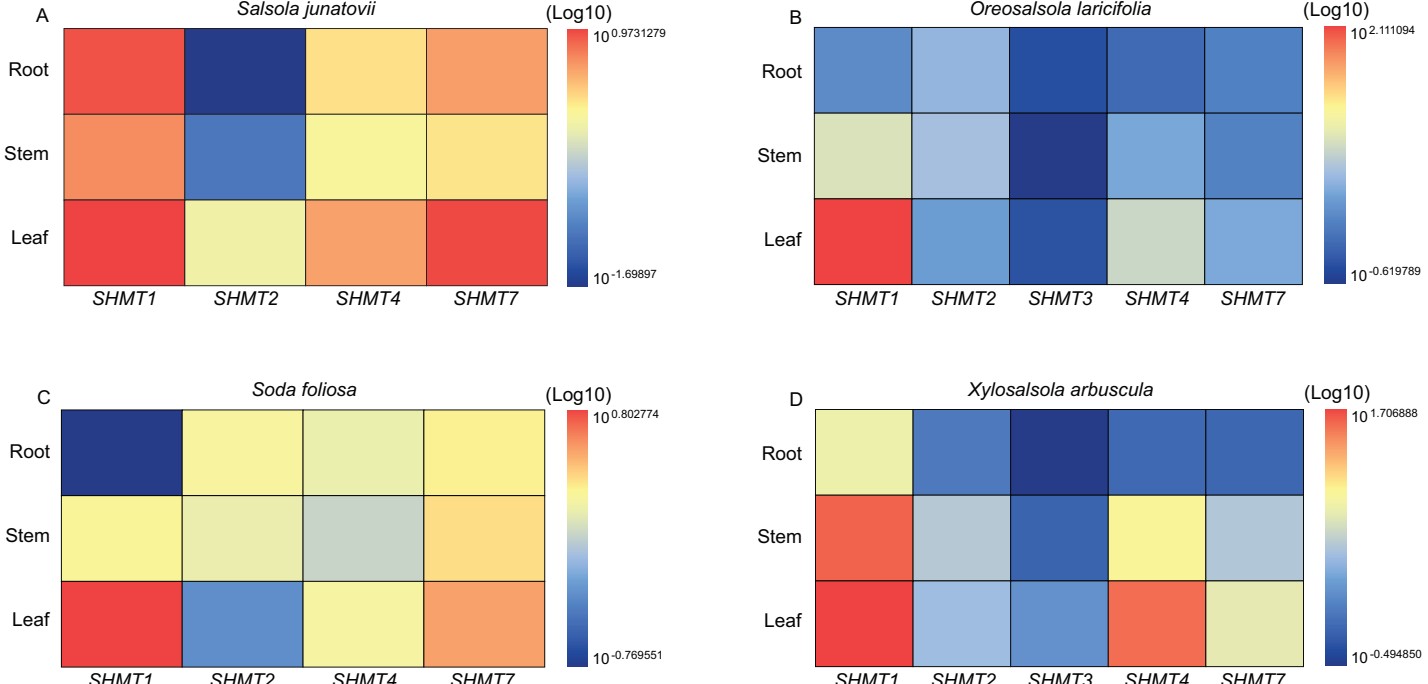

**Figure 4 Relative expression levels of *SHMT* members in roots, stems, and leaves of four Salsoleae species.** The relative expression levels were calculated using the $2^{-\Delta\Delta CT}$ method, and the data were log-transformed (log10) and standardized for visual comparison. The color scale indicates expression levels, with blue representing lower expression and red representing higher expression.

(Molluginaceae), and *Neurachne* (Poaceae) (Figs. 5 and S3, Table. S2). The results indicated that, except for *MpenSHMT1*, *FkocSHMT4*, and *McerSHMT7*, which were not detected in *Mollugo pentaphylla*, *Flaveria kochiana*, and *M. cerviana*, respectively, all other species exhibited the expression of *SHMT1*, *SHMT4*, and *SHMT7*. Additionally, *SHMT2*, *SHMT3*, and *SHMT6* were detected in some species. Although *SHMT* genes were widely distributed across various plant groups, significant differences existed in the number of gene members and their expression levels. Notably, the types and expression patterns of *SHMT* members varied among different species within these genera. In general, the expression levels of *SHMT6* and *SHMT7* were relatively low compared to other *SHMT* members, while *SHMT2*, *SHMT3*, and *SHMT4* exhibited relatively higher expression levels in specific groups. Furthermore, *SHMT1* was the most predominantly expressed member in most species and showed higher expression levels in $C_2$ and $C_3$ plants compared to $C_4$ plants.

## Cis-acting elements and transcription factor binding site analysis

In our study, *SHMT1* was identified as the predominantly expressed *SHMT*. Consequently, we conducted a detailed analysis of the 2,000 bp upstream promoter region of *SHMT1* across four Salsoleae species. A total of 40 key cis-acting elements was detected, including 12 light-responsive elements, eight growth and development-related elements, 11 hormone-responsive elements, and nine stress-responsive elements (Fig. 6A). The types of cis-acting elements presented in the *SHMT1* promoter region varied among these four

**Peng et al. (2025),** *PeerJ*, **DOI 10.7717/peerj.19978**      14/25

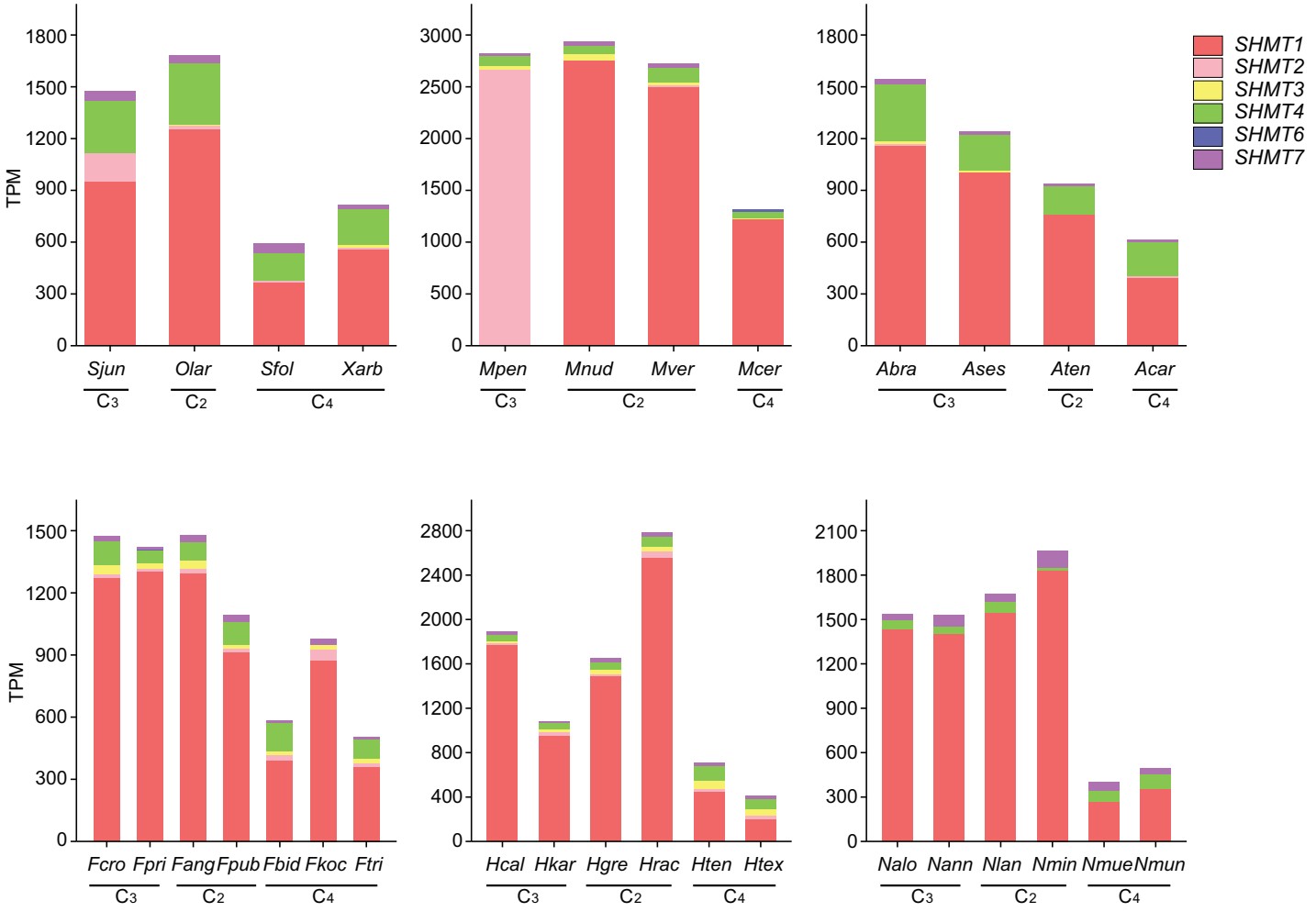

**Figure 5 The expression levels of *SHMTs* in leaves across six different plant groups.** With distinct colors representing different *SHMT* members. Transcript levels are presented as transcripts per million (TPM). All data are derived from published RNA-seq studies. species abbreviations are as follows: Sjun (*Salsola junatovii*), Sfol (*Soda foliosa*), Olar (*Oreosalsola laricifolia*), Xarb (*Xylosalsola arbuscula*), Fcro (*Flaveria cronquistii*), Fpri (*Flaveria pringlei*), Fang (*Flaveria angustifolia*), Fpub (*Flaveria pubescens*), Fbid (*Flaveria bidentis*), Fkoc (*Flaveria kochiana*), Ftri (*Flaveria tri-nervia*), Hcal (*Heliotropium calcicola*), Hkar (*Heliotropium karwinskyi*), Hgre (*Heliotropium greggii*), Hrac (*Heliotropium racemosum*), Hten (*Heliotropium tenuifolium*), Htex (*Heliotropium texanum*), Mpen (*Mollugo pentaphylla*), Mnud (*Mollugo nudicaulis*), Mver (*Mollugo verticillata*), Mcer (*Mollugo cerviana*), Ases (*Alternanthera sessilis*), Abra (*Alternanthera brasiliana*), Aten (*Alternanthera tenella*), Acar (*Alternanthera caracasana*), Nalo (*Neurachne alopecuroideae*), Nann (*Neurachne annularis*), Nlan (*Neurachne lanigera*), Nmin (*Neurachne minor*), Nmue (*Neurachne muelleri*), Nmun (*Neurachne munroi*); data sources and other relevant information are provided in Table S5.

species. Specifically, *S. junatovii* (C$_3$) exhibited the highest number of distinct cis-acting element types, with 29 identified, while *O. laricifolia* (C$_2$) had the fewest, with only 19. Further analysis revealed that the frequency of occurrence of different cis-acting elements also varied among these species. For instance, the Sp1 element appeared five times in *S. junatovii*, whereas its occurrence was lower in the other species. The Box-4 element occurred most frequently in *O. laricifolia*, with six occurrences. We further analyzed the positions of cis-acting elements within the *SHMT1* promoter regions across various species. Our findings revealed significant variations in both the quantity of elements and

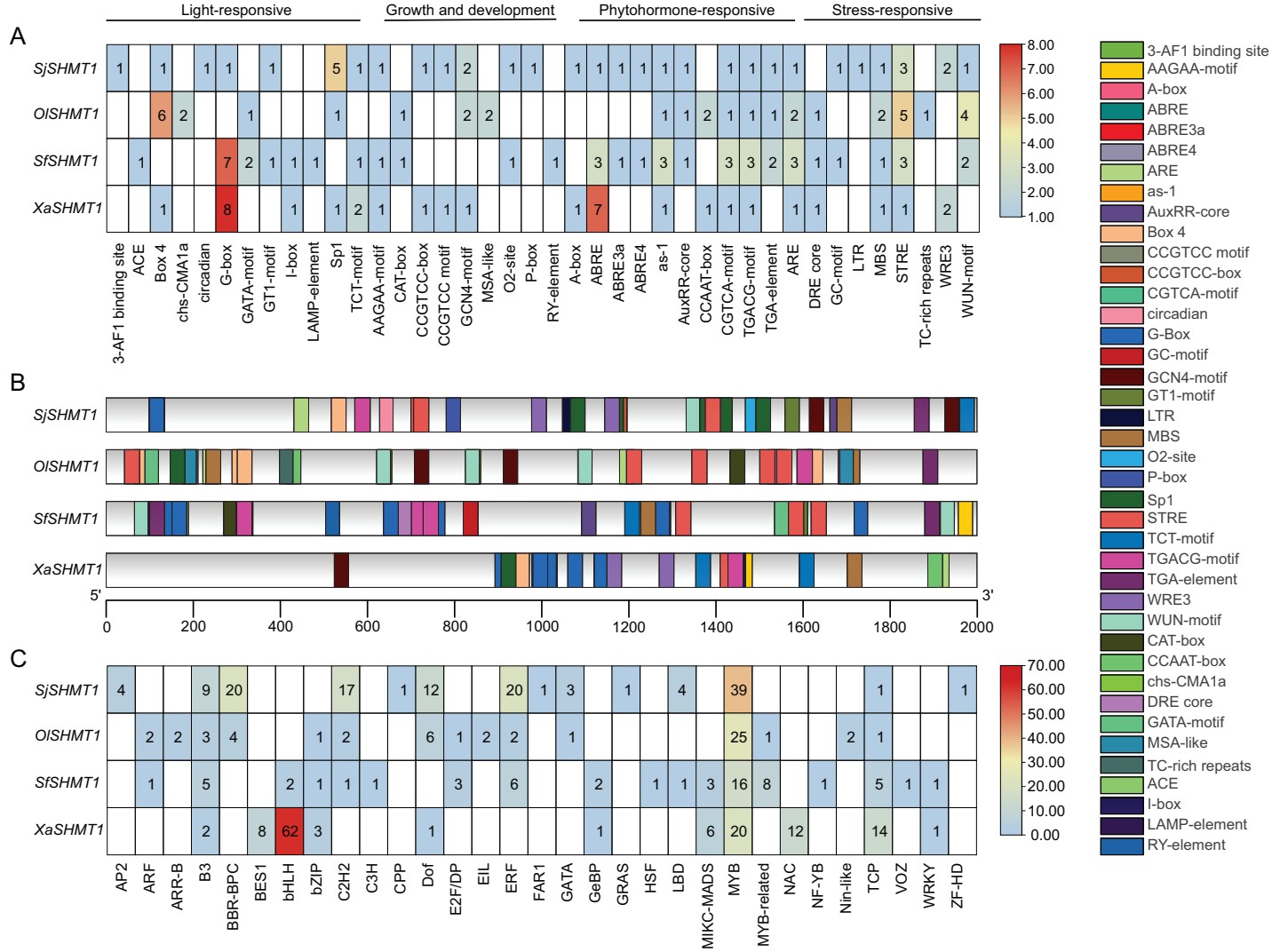

**Figure 6  Prediction of cis-elements and bounding transcription factors on the *SHMT* promoters of four Salsoleae species.** (A) The types and numbers of cis-elements on the promoters. (B) The locations of cis-elements on the genes. (C) Putative transcription factors binding to the *SHMT* promoters.

their specific locations within these regions among different species. For example, *OlSHMT1* exhibited a higher concentration of cis-acting elements near the transcription start site, whereas *XaSHMT1* displayed fewer elements that were more dispersed (Fig. 6B).

Additionally, we predicted the transcription factor binding sites (Fig. 6C). The results indicated that *X. arbuscula* possessed only 11 transcription factor binding sites in its *SHMT1* promoter, with the BasicHelix-Loop-Helices (BHLH) binding site being particularly prevalent, occurring a total of 62 times. In contrast, the *SHMT1* promoter of *S. foliosa* exhibited the greatest diversity, identifying 18 different types of transcription factor binding sites. Our analysis further revealed that v-myb avian myeloblastosis viral oncogene homolog (MYB) binding sites were abundant in the *SHMT1* promoters across all species. Moreover, certain transcription factor binding sites displayed distinct species

specificity, for instance, APETALA2 (AP2), Cysteine-rich Polycomb-like (CPP), and Far-red Impaired Response 1 (FAR1) binding sites were detected solely in the *SHMT1* promoter of *S. junatovii*. In summary, these findings reveal significant species-specific differences in the types, numbers, and distribution of cis-acting elements and transcription factor binding sites within the *SHMT1* promoter among different Salsoleae species, suggesting that these elements may play unique roles in environmental adaptation and the regulation of photosynthesis.

## DISCUSSION

SHMT catalyzes the reversible interconversion of glycine and serine with GDC, and a crucial enzyme in the cell-carbon metabolic pathway (*Schirch, 1982*; *Hanson, Gage & Shachar-Hill, 2000*; *Appaji Rao et al., 2003*; *Zhang et al., 2010*). Genes encoding the *SHMT* family have been identified in various higher plants, such as five in rice (*Pan et al., 2024*); seven in each *A. thaliana* (*Zhang et al., 2010*) and tomato (*Liu et al., 2022*), 9 in *P. trichocarpa* (*Li & Cheng, 2020*), 15 in alfalfa (*Gao et al., 2024*), 18 in soybean (*Lakhssassi et al., 2020*). In this study, these species contained the close identified *SHMT* number in rice but less than the most published higher plants. We identified 4 *SHMTs* in both *S. junatovii* and *S. foliosa*, and they are distributed across two chromosomes, respectively. Additionally, 5 *SHMTs* were identified in both *O. laricifolia* and *X. arbuscula*, and they are distributed on three chromosomes, respectively (Table 1, Fig. S2). The number of *SHTMs* appears to be correlated with gene duplication event and the number of whole-genome duplications. *SHMTs* in cucumber (*Gao et al., 2022*), alfafa (*Gao et al., 2024*), and soybean (*Lakhssassi et al., 2020*) participate in a fragment duplication event, respectively. The soybean genome contains 18 *SHMTs* and has likely undergone a larger-scale genome replication event (*Gao et al., 2024*). However, there was no fragment duplication events, and no gene is involved in tandem duplication in four Salsoleae species (Fig. 3), which maybe cause the less *SHMT* members in each species.

To investigate the relationship between the *SHMT* members of four Salsoleae species and other species including *A. thaliana*, *O. sativa*, *G. max*, *S. lycopersicum*, *P. trichocarpa*, *C. sativus*, *B. vulgaris*, we classified these *SHMTs* into four clades, which generally corresponded to their subcellular localizations, with the exception of in Class II (Fig. 2). In Class III, consistent with previous studies (*Gao et al., 2022*, *2024*; *Pan et al., 2024*), we observed the absence of Class III *SHMT* in rice. Additionally, Class III *SHMT* was also absent in *S. junatovii* and *S. foliosa*. However, the predicted chloroplast-localized *SjSHMT4* from *S. junatovii* and *SfSHMT4* from *S. foliosa* were present in Class II, suggesting that gene duplication may have compensated for the loss of Class III. Furthermore, *OsSHMT3* from rice, originally classified within the cytoplasmic-localized group, has been found to be localized in chloroplasts within rice protoplasts (*Pan et al., 2024*). And there were no cytoplasmic-localized *SHMT* was predicted in these four Salsoleae species. The overexpression of *PtSHMT2* from *Populus* promotes growth by enhancing biomass production and the release of sugars such as glucose and xylose (*Zhang et al., 2019a*). Whether a specific *SHMT* loss event occurred in these Salsoleae species still requires extensive experimental validation. Furthermore, genes belonging to the same class

exhibited high similarity in both gene structure and motif numbers, with the exception of *OlSHMT3* and *XaSHMT3* in terms of gene structure (Fig. 1). Collinearity analysis revealed that *OlSHMT3* is collinear with *SHMT3* from *Beta*, but no collinear relationship was found between *XaSHMT3* and *SHMT3* from Beta (Fig. 3, Table S7), indicating that these two genes have undergone evolutionary and adaptive structural changes. Differences in exon numbers were observed across different classes rather than in between species. The range of exon numbers were from 4 to 15 (Fig. 1). The diverse gene structures of the *SHMTs* in the four Salsoleae species may. result from an evolutionary process characterized by intron loss or gain (*Ding et al., 2014*; *Lakhssassi et al., 2019*).

To investigate the roles of *SHMTs* during the growth and development of four Salsoleae species, this study analyzed the expression levels of *SHMTs* in leaves, stem and root, revealing that these *SHMTs* are expressed across all tissues examined. Each gene exhibited distinct tissue expression patterns. Notably, the mitochondrial-localized *SHMT1* from the four Salsoleae species showed higher gene expression levels in leaves compared to stem and root (Fig. 4). This finding aligns with previous reports in *Arabidopsis* (*Moreno, Martín & Castresana, 2005*), alfalfa (*Gao et al., 2024*), soybean (*Lakhssassi et al., 2019*). Conversely, the mitochondrial-localized *SHMT2* demonstrated different expression patterns among in the Salsoleae species. In *O. laricifolia* (Fig. 4B) and *S. foliosa* (Fig. 4C), *SHMT2* expression was higher in stem and roots than in leaves. In contrast, For *S. junatovii* (Fig. 4A) and *X. arbuscula* (Fig. 4D), exhibited higher *SHMT2* expression in stem and leaves compared to root. Moreover, no collinear relationship was observed between *SHMT1* and *XaSHMT1* across Salsoleae species, *Beta*, and *Arabidopsis* (Fig. 3, Table S7). The diversification of predominant tissue expression pattern between mitochondrial-localized *SHMT1* and *SHMT2* in these Salsoleae species may indicate function diversification of these genes (*Voll et al., 2006*). Furthermore, while mitochondrial-localized *SHMT1* and *SHMT2* in *Arabidopsis* exhibit different tissue expression patterns and are not functionally redundant (*Voll et al., 2006*), the transcript accumulation of *AtSHMT4* in *Arabidopsis* is restricted to the roots of seedlings (*Moreno, Martín & Castresana, 2005*). In contrast, chloroplast-localized *SHMT4* in the four Salsoleae species and *SHMT3* in *X. arbuscula* demonstrated higher gene expression in leaes thanin stem and root (Fig. 4). In *O. laricifolia*, *SHMT3* exhibited higher gene expression in roots (Fig. 4B). Although both *SHMT4* and *SHMT3* are localized in chloroplasts, the expression level of *SHMT4* is greater than that of *SHMT3* in *X. arbuscula* (Fig. 4D) and *O. laricifolia* (Fig. 4B). Chloroplastic *SHMT* plays a crucial role in photoreception and the biosynthesis related to one carbon metabolism (*Hanson, Gage & Shachar-Hill, 2000*; *Zhang et al., 2010*).

In $C_4$ species, the repression of the Rubisco oxygenation reaction and the absence of toxic byproducts result in the low expression of key genes associated with most enzymes in the photorespiratory cycle (*Mallmann et al., 2014*). We utilized the newly sequenced leaf transcriptome of four Salsoleae species, along with published leaf transcriptome data from five genera were used to examine the expression of *SHMTs* along the emergence of $C_4$ species. Compared with other *SHMTs*, only mitochondrial-localized *SHMT1* exhibited significantly high transcript abundance in leaves, with the exception of *M. pentaphylla*,

which did not express *SHMT1*. Furthermore, the expression of *SHMT1* in $C_4$ species was lower compared to that in $C_3$ and $C_2$ species. Notably, the transcript levels of *SHMT1* in $C_2$ plants remained constant or were even higher than those observed in $C_3$ plants (Fig. 5). *AtSHMT1* and *AtSHMT4* from *Arabidopsis* are regulated by the circadian clock, aligns with their role in photorespiration (*McClung et al., 2000*). However, the expression of *SHMT4* in $C_4$ species was inconsistent compared to that in $C_3$ and $C_2$ species within these genera (Fig. 5). *AtSHMT1*, the *SHMT* coding gene from *Arabidopsis*, plays a crucial role in the photorespiratory cycle (*Voll et al., 2006*). It is concluded that the low expression of the leaf preferential mitochondrial-localized *SHMT1* has evolved during the evolution of $C_4$ photosynthesis. The cis-acting element and the transcription factors bound to the *SHMT1* promoters in four Salsoleae species were predicted (Fig. 6). Our analysis revealed that the *SHMT1* promoter regions in these species contained various cis-acting elements associated with light response, growth and development, phytohormone response, and stress response (Fig. 6A). This indicates that the function of *SHMT1* in these Salsoleae species may encompass these four aspects. The MYB factors were predicted to be the most significant transcription factors binding to *SHMT1* in three Salsoleae species, and the second most significant in *X. arbuscula* (Fig. 6C). Further verification through additional experiments is needed to confirm the binding of MYB factors to *SHMT1* in Salsoleae. MYB factors are implicated in various aspects of $C_4$ photosynthesis, including cell/division/size (*Rao et al., 2016*), bundle sheath wall formation (*Rao et al., 2016*), sulfur metabolism, glucosinolate biosynthesis (*Aubry et al., 2014*) and achieving cell specific expression (*Dickinson et al., 2020*; *2023*).

## CONCLUSIONS

In this study, we systematically analyzed the *SHMT* gene family across four Salsoleae species: $C_3$ species *S. junatovii*, $C_3$-$C_4$ intermediate *O. laricifolia*, and $C_4$ species *X. arbuscula* and *S. foliosa*. We identified four or five *SHMTs* in each species, with no instances of fragment duplication detected, which may account for the relatively low number of family members. The number of exons in *SHMTs* varied from four to fifteen, and these genes were classified into four distinct phylogenetic groups. *SHMTs* within each phylogenetic group shared similar exon-intron structures and conserved motif compositions. Expression analysis revealed that, although *SHMTs* are not strictly organ-specific, the mitochondrial-localized and leaf-preferential *SHMT1* exhibited significantly lower expression levels in $C_4$ species compared to those in $C_3$ and $C_3$-$C_4$ intermediate species. This observation suggests that the regulation of this gene may have evolved in the $C_4$ photosynthesis. Furthermore, cis-acting element analysis predicted that MYB transcription factors may serve as key regulators of *SHMT1* in several Salsoleae species. Overall, these findings establish a foundation for further exploration of the function and evolution of the *SHMT* gene family in Salsoleae, contributing to a deeper understanding of the molecular mechanisms underlying the evolution of photosynthetic pathways.

### Funding

This work was supported by the Tianshan Talent Training Program
(No. 2023TSYCCX0090), the Shanghai Cooperation Organization Partnership and
International Technology Cooperation Plan of Science and Technology Projects
(No. 2022E01033), and the National Science Foundation of China (No. 42271072). The
funders had no role in study design, data collection and analysis, decision to publish, or
preparation of the manuscript.

### Grant Disclosures

The following grant information was disclosed by the authors:
Tianshan Talent Training Program: 2023TSYCCX0090.
Shanghai Cooperation Organization Partnership and International Technology
Cooperation Plan of Science and Technology Projects: 2022E01033.
National Science Foundation of China: No. 42271072.

### Competing Interests

The authors declare that they have no competing interests.

### Author Contributions

- Peng Peng conceived and designed the experiments, performed the experiments,
  analyzed the data, prepared figures and/or tables, and approved the final draft.
- Qian Qin performed the experiments, prepared figures and/or tables, and approved the
  final draft.
- Guzailinuer Kuerban conceived and designed the experiments, prepared figures and/or
  tables, and approved the final draft.
- Ting Peng performed the experiments, prepared figures and/or tables, and approved the
  final draft.
- Mao Wang conceived and designed the experiments, authored or reviewed drafts of the
  article, and approved the final draft.
- Zhibin Wen conceived and designed the experiments, authored or reviewed drafts of the
  article, and approved the final draft.

### DNA Deposition

The following information was supplied regarding the deposition of DNA sequences:
The newly sequenced transcriptome raw data for the four Salsoleae species are available
at NCBI: SAMN47206123 (*Salsola junatovii*), SAMN47206125 (*Oreosalsola laricifolia*),
SAMN47206127 (*Salsola foliosa*/*Soda foliosa*), and SAMN47206129 (*Xylosalsola
arbuscula*).

### Data Availability

The RNA-Seq data of the four newly sequenced species of the Salsoleae tribe are available at NCBI: SRR32571875 (*Salsola junatovii*); SRR32571874 (*Oreosalsola laricifolia*); SRR32571873 (*Soda foliosa*); SRR32571872 (*Xylosalsola arbuscula*).

The gene family protein sequences, and other RNA-seq data summary information are available in the Supplemental File.

## Supplemental Information

Supplemental information for this article can be found online at http://dx.doi.org/10.7717/peerj.19978#supplemental-information.

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
