# Peer review of "Genome-wide identification of SHMT family genes in C3, C3-C4, and C4 Salsoleae s.l. species"

_PeerJ, doi:10.7717/peerj.19978_

## Round 0.1 · original submission · Major Revisions

We received comments from two independent Reviewers. I urge the Authors to provide throrough responses to them.

As for myself, I find the manuscript interesting, but needing much clarification. For instance, the Abstract starts with the sentence about C4 evolution - please explain what the C4 is. Also, what does "there is a lack of comprehensive bioinformatics investigation on the different photosynthetic species" mean? By photosynthetic species the Authors mean organisms? If so, there's plenty of studies on green plants etc. Please be more specific in here but also across the manuscript.

Why the Authors chose Salsoleae species for the analysis?

Remove the root from unrooted tree. I understand that presented form of the tree is convinient but the presence of the root is misleading. Anyway - what substitution model was used for the phylogenetic tree construction, and how was that chosen?

Just an aesthetic comment, but potentially cleaning up the figures and improving readability - when showing rectangles, like those on fig 1, use solid colors instead or gradients.

Please remove non-informative columns from table, like "Signal peptide likelihood' from the Table 2 - since all cases have 'NO' there's no point in providing this information in the table. Also, I'm not convinced that percentage of secondary structures is informative. Rather, the Authors might want to compare 3D structural models, or analyze the presence of additional protein domains. It is hard for the reader to assess the meaning of strand amount differences.

Please provide NCBI gene identifiers in table 1. Ideally, provide also UNIPROT identfiers for proteins encoded. DEspite protein sequences provided in the supplement, please provide nucleotide sequences used for cis-regulatory element analyses.

I'd like also to inspire the Authors to work on the reporting in order to make it more self-explanatory and clear for the readers.

**Language Note:** The review process has identified that the English language must be improved. PeerJ can provide language editing services - please contact us at [email protected] for pricing (be sure to provide your manuscript number and title). Alternatively, you should make your own arrangements to improve the language quality and provide details in your response letter. – PeerJ Staff

·

Basic reporting

The authors classified the four Salsoleae species, alongside the species obtained from databases (see Figure 5 and Table S5), into three photosynthesis types: C3, C4, and C3-C4 intermediate. Is there clear evidence to support the photosynthesis types of these species with regard to physiological or anatomical aspects of the leaves? For example, leaf delta 13C values and leaf anatomy are known to differ significantly between C3 and C4. I searched the delta 13C values of these species in a database (Cornwell WK, et al. 2017. A global dataset of leaf Δ13C values. Zenodo. DOI: 10.5281/zenodo.569501); however, I could not find any for these species. In the revised manuscript, the authors should clearly provide evidence to support the photosynthesis types of these species.

Experimental design

-

Validity of the findings

For qRT-PCR, the expression levels of SHMT genes were likely measured relative to those of actin genes (see Raw data-Table), although this is not explicitly stated in the Materials and Methods section (see line 196-). The authors must clearly state that the actin genes were used as a control. Moreover, the primers used for actin genes for each species must be provided alongside the primers for the SHMT genes shown in Table S6.

In Figure 4, the expression levels of the SHMT gene members of different species are expressed as the SHMT gene expression level of Salsola junatovii roots as a control (see lines 220-221 in the text). In the text, the authors compare the expression levels of SHMT gene members between the plant tissues, not between the different species (see lines 342-354). Therefore, for example, the expression levels of SHMT gene members in plant tissues would be more appropriately expressed as the SHMT gene expression level of the root for each species.

Additional comments

Line 36, Abstract: Salsola junarovii, “junatovii” is correct?

Line 103: “…expressed in shoots and roots”. A reference would be needed after this sentence.

Line 203: The unit Lux is not appropriate for plant research. It should be expressed in terms of photosynthetic photon flux density instead.

Line 578: Lakhssassi, N, Piya S,…A new paragraph should start before this sentence.

Figure 5: The species abbreviations should be included in the figure legend itself, rather than in a supplementary table.

Reviewer 2 ·

Basic reporting

The submitted work is interesting, but, in my opinion, the manuscript presents some issues mainly in the Materials and Methods, Figures, and in the text, as well as other things, that need to be checked to make this work suitable for publication.

Text comments
Line 49. Please correct “Xa. arbuscula” with “X. arbuscula”. The name of this species should be in italics.
Line 197. Given that foliosa is classified within the genus Soda, there is no possibility of confusion with the genus Salsola. Please correct “So. foliosa” with “S. foliosa”. Please review the entire text.
Line 220. Please correct “Sciences. (Zhang et al., 2019)” with “Sciences (Zhang et al., 2019).”
Line 235. Authors are reminded that tables and figures should appear consecutively in the text. The first table to appear is Table S5.
Lines 253 and 375. Please correct “2000” with “2,000”
Line 271. Please correct “68405.64” with “68,405.64”
Line 272. Please correct “50244.21” with “50,244.21”
Line 360. In this line, Figure 5 and Figure S5 are cited. I think that Table S5 should be cited instead of Figure S5, which does not exist.
Lines 304 and 423. Please correct “Solanum lycopersicum” with “S. lycopersicum”
Line 433. Please, the word Populus refers to a genus and should be in italics.
Line 472. In my opinion, the reference to Table S6 should be removed

Figure comments:
The quality of Figures 1 and 3 appears to be low, possibly due to the PDF conversion process. In any case, it would be advisable to review and, if necessary, replace them with higher-resolution versions.

Figure 3a. The yellow color indicating the chromosomes is not clearly visible. Please change it to another color that is more visible.

Table comments:
Table 1. The molecular weight values are not formatted with commas to separate thousands. In Molecular Weight and PI, the values ​​have a space after the point that is left over.

Table 2: In Alpha helix, extended strand, beta turn, and Random coil, the values ​​have a space after the point that is left over.

Experimental design

The experimental design is adequate in all respects. The authors should comment or reference the seed sterilization methods on line 199 (section "RNA extraction and Quantitative real-time PCR"). At the end of this section, the authors should cite the primer table. The authors should explain the reason for collecting the samples at 11:00 AM; furthermore, this information is irrelevant without knowing what time the lights are turned on or off.

Validity of the findings

The results and conclusions are well supported by the results obtained.

---

## Round 0.2 · accepted · Accept

Both Reviewers and I are satisfied with the changes made by the Authors since the last revision, and the manuscript may be published in its current form.

·

Basic reporting

no comment

Experimental design

no comment

Validity of the findings

no comment

Additional comments

Line 488: A typographical error needs to be corrected: "leaes thanin"->"leaves than in".

Reviewer 2 ·

Basic reporting

The authors have adequately addressed the comments raised during the initial review. As a result, the manuscript has undergone substantial improvement

Experimental design

The authors have adequately addressed the comments raised during the initial review. As a result, the manuscript has undergone substantial improvement

Validity of the findings

The results and conclusions are well supported by the results obtained.